# Properties of Multidrug-Resistant Mutants Derived from Heterologous Expression Chassis Strain *Streptomyces albidoflavus* J1074

**DOI:** 10.3390/microorganisms11051176

**Published:** 2023-04-30

**Authors:** Borys Dolya, Olena Hryhorieva, Khrystyna Sorochynska, Maria Lopatniuk, Iryna Ostash, Vasylyna-Marta Tseduliak, Eva Baggesgaard Sterndorff, Tue Sparholt Jørgensen, Tetiana Gren, Yuriy Dacyuk, Tilmann Weber, Andriy Luzhetskyy, Victor Fedorenko, Bohdan Ostash

**Affiliations:** 1Department of Genetics and Biotechnology, Ivan Franko National University of Lviv, 79005 Lviv, Ukraineviktor.fedorenko@lnu.edu.ua (V.F.); 2Department of Pharmacy, Saarland University, 66123 Saarbrucken, Germany; 3The Novo Nordisk Foundation Center for Biosustainability, Technical University of Denmark, Kemitorvet, 2800 Kongens Lyngby, Denmarktiwe@biosustain.dtu.dk (T.W.); 4Department of Mineral Geology and Geophysics, Ivan Franko National University of Lviv, 79005 Lviv, Ukraine

**Keywords:** *Streptomyces albidoflavus* J1074, rifampicin resistance, *rpoB*, specialized metabolites

## Abstract

*Streptomyces albidoflavus* J1074 is a popular platform to discover novel natural products via the expression of heterologous biosynthetic gene clusters (BGCs). There is keen interest in improving the ability of this platform to overexpress BGCs and, consequently, enable the purification of specialized metabolites. Mutations within gene *rpoB* for the β-subunit of RNA polymerase are known to increase rifampicin resistance and augment the metabolic capabilities of streptomycetes. Yet, the effects of *rpoB* mutations on J1074 remained unstudied, and we decided to address this issue. A target collection of strains that we studied carried spontaneous *rpoB* mutations introduced in the background of the other drug resistance mutations. The antibiotic resistance spectra, growth, and specialized metabolism of the resulting mutants were interrogated using a set of microbiological and analytical approaches. We isolated 14 different *rpoB* mutants showing various degrees of rifampicin resistance; one of them (S433W) was isolated for the first time in actinomycetes. The *rpoB* mutations had a major effect on antibiotic production by J1074, as evident from bioassays and LC-MS data. Our data support the idea that *rpoB* mutations are useful tools to enhance the ability of J1074 to produce specialized metabolites.

## 1. Introduction

*Streptomyces albidoflavus* J1074 (also known as *S. albus* J1074) is one of the most popular and genetically tractable platform strains widely used for drug discovery purposes [1]. The strain has been very successfully used for the expression of (meta)genomic libraries [2] and individual cosmids [3], which led to isolation of a number of specialized metabolites (SMs) and its use for the incorporation of nonproteinogenic amino acids into ribosomally produced peptides [4]. A low SM yield is a known shortcoming for all currently available streptomycete chassis strains, fueling the interest in strain improvement efforts. In this regard, a lot has been done to tailor the genetic makeup of J1074 for heterologous expression purposes. The major achievements include the development of vectors, reporters, and other genetic ‘’biobricks’’ (terminators, RBSs, etc.) of different types [5,6]; genome minimization [7]; and the introduction of beneficial genes and mutations [8]. We have previously described a set of genetically engineered J1074 mutants carrying point mutations within the *rpsL* gene encoding the ribosomal protein S12 [9,10]. Such mutations are known to enhance the production of SMs by streptomycetes [11]. The aforementioned J1074 *rpsL* mutants fall into two classes. In the first class, the wild type *rpsL* allele was substituted with a mutant one (so called “pure” mutants). In the second class, the *rpsL* mutant allele was introduced on an integrative plasmid into *S. albidoflavus* in the presence of a wild type *rpsL* gene (merodiploids). We used *rpsL*^R94G^-carrying strains as a starting point to sequentially introduce spontaneous mutations of resistance to streptomycin, lincomycin, erythromycin, and rifampicin, with an ultimate aim to improve the ability of *S. albidoflavus* to produce SMs [12]. A summary of all generated spontaneous mutants is given in Figure 1.

Within the set of strains depicted in Figure 1, we recently characterized streptomycin and lincomycin resistance mutations in a lineage of pure mutants [13] and described the levels of rifampicin resistance and the identity of mutations within the *rpoB* gene (coding for the β-subunit of RNA polymerase known to cause Rif^r^) of KO-1401—1417 strains [12]. Yet, the effects of erythromycin and rifampicin resistance (Rif^r^) mutations on morphology, the specialized metabolome, and susceptibility to the other drugs of J1074 remained unstudied. In this work, we focus on the characterization of properties of Rif^r^ mutants raised on the basis of *rpsL* pure (KO-1305) and merodiploid (KO-1307) mutants (see Figure 1). While our mutagenesis resulted in more than a dozen point mutations in rpoB that have already been described in the literature, one is described here for the first time. Some of the mutants display altered antibiotic resistance spectra and an enhanced ability to produce SMs under certain conditions. Analysis of genomes of a few Rif^r^ mutants did not reveal any unanticipated changes that would contribute to their SM-overproducing phenotypes. This implies that *rpoB,* in combination with the previously introduced *rpsL*/*rsmG*/*xnr_2147* mutations, is the reason for the enhanced specialized metabolism of the studied strains.

## 2. Materials and Methods

### 2.1. General Experimental Procedures

The *S. albidoflavus* strains studied in this work are described in [12]; see also Figure 1. *Staphylococcus aureus* 209P, *Bacillus cereus* ATCC 19637, and *Debaryomyces hansenii* VKM Y-9 were used to assay the antibiotic activity of *S. albidoflavus* strains as described in [13]. *E. coli* DH5α was used for routine cloning purposes. *E. coli* ET12567 (pUZ8002) was used to transfer cosmid pOJara-harboring aranciamycin BGC [10] and other plasmids according to the mating protocol described in [13]. Solid media SG2, SFM, R5S (see the recipe below), TSA, MM, and GYM [14] were used in the agar plug assays of antibiotic activity of *S. albidoflavus* strains. Tryptic soy broth (TSB; Merck KgA, USA) was used to grow J1074 biomass for DNA isolation and for antibiotic resistance assays. For submerged production of endogenous SMs, *S. albidoflavus* strains were grown in liquid media SG2 [14] and R5S (g/L: sucrose—103, K_2_SO_4_—0.25, MgCl_2_ × 6H_2_O—10.12, glucose—10, casamino acids—0.1, yeast extract—5, TES—5.73, and microelements solution [15]—2 mL). For aranciamycin production, pOJara^+^ strains were grown in liquid medium SG1 [14]. To determine antibiotic resistance profiles of *S. albidoflavus* strains, we used either commercially available antibiotic disks (Himedia Ltd., Mumbai, India) or impregnated pure paper disks (Ø 6.35 mm; Schleicher and Schuell Co., Keene, NH, USA) with a certain amount of antibiotic, as indicated below in Section 3. We followed the established procedure of the antibiotic disk diffusion assay described in [16]. Resazurin assay, as described in [17], was also employed to determine minimal inhibitory concentrations (MICs) of erythromycin for strains KO-1305 and KO-1307. Unless otherwise stated, all media components were from Sanimed Ltd. (Kharkiv, Ukraine), molecular biology enzymes were from Thermo, and other reagents and antibiotics were purchased from Carl Roth GmbH (Karlsruhe, Germany).

### 2.2. Analysis of Specialized Metabolome of S. albidoflavus Strains under Submerged Conditions

A piece of sporulated lawn (approx. 2 sq. cm) of J1074 strains (grown on SFM agar for 7 days at 30 °C) was used to inoculate a 300 mL flask containing 40 glass beads (Ø 5 mm, Sigma-Aldrich, St. Louis, MO, USA) and 35 mL of fermentation medium. The inoculated flasks were incubated on an orbital shaker (180 rpm, 30 °C) for 120 h. A volume of 15 mL of resulting fermentation broth was extracted with an equal volume of ethyl acetate. Organic phase was evaporated in vacuo, and dry residue from a 5 mL fraction was dissolved in 100 µL of methanol. A volume of 12 µL of the methanol solution was used in disk assay of antibiotic activity, and 1 µL was subjected to LC-MS analysis, as described in [10]. MS analysis was carried out both in negative and positive ionization modes. Spectrophotometric quantification of aranciamycin production by recombinant *S. albidoflavus* strains followed the protocol described in [10].

### 2.3. Sequencing and Analysis of Genomes of Selected Mutants

Genomic DNA from strains KO-1305, KO-1307, KO-1403, KO-1407, KO-1408, and KO-1412 (see Figure 1) was extracted from 24-hour-old cultures following the described procedure [18]. DNA concentrations and quality were determined using Trinean Xpose (Gentbrugge, Belgium). Illumina TruSeq DNA PCR-Free and NEB Next Ultra II DNA Library Prep (Cat No E7645) Kits were used to make WGS libraries. The Illumina library was sequenced on Novaseq platform. We verified *rpoB* mutations via amplification of the gene (*xnr_3712*) with primers Xnr3712_xbaI_up AAATCTAGAGTCCGAGCCCTCGGAAGG, and Xnr3712_mfeI_up AAACAATTGGGTCAGACCTCTTCGACG and subsequent Sanger sequencing.

Raw data of aforementioned *S. albidoflavus* genomes, reference sequence of J1074, as well as an Excel table listing the called variants and Appendix A have been deposited in the J1074 genomics database maintained by Lviv University, available at https://biotools.online/media/ (accessed on 1 January 2023). At the quality control stage, the sequence reads were examined for overall quality and presence of Illumina adapters with FastQC [19]. Low-quality reads were either trimmed or filtered out using Trimmomatic version 0.36 [20]. Sequencing reads were aligned to reference the J1074 genome (NCBI accession number CP004370) with Bowtie2 version 2.2.5 [21]. Detection of SNVs and indels was performed with ReadXplorer [22]. Illumina coverage was from 55- to 165-fold for all strains (detailed list with average coverage for assembled genomes can be found on the aforementioned webpage in xlsx files in the SNP folder for coverage on variant calling).

### 2.4. Cloning of rpoB Alleles

Primers Xnr3712_xbaI_up and Xnr3712_mfeI_rp were used to amplify gene *rpoB* from the wild-type strain (SAM2) and mutants KO-1403 (allele *rpoB*^S433W^) and KO1407 (*rpoB*^R440C^). The resulting 3.4 kb amplicons were treated with restriction endonucleases XbaI and MfeI and ligated to XbaI-EcoRI-digested integrative expression vector pTES [23]. Recombinant plasmids harboring wild-type *rpoB*, *rpoB*^S433W^, and *rpoB*^R440C^ were labeled as pTESrpoB-WT, pTESrpoB-1403, and pTESrpoB-1407, respectively.

## 3. Results

### 3.1. Generation and Analysis of Erythromycin Resistant Strains KO-1305 and KO-1307

By plating spore suspensions (approx. 4 × 10^8^ CFU/mL) of KO-1304 and KO-1301 strains onto GYM agar supplemented with 50 µg/mL of erythromycin, we were able to isolate three and four erythromycin-resistant (Ery^r^) mutants of KO-1304 and KO-1306, respectively. The mutants did not differ from their parents in colony morphology or growth rate (see also Appendix A). In agar plug bioassays against S. aureus 209P the GYM-grown Ery^r^ mutants derived from KO-1304 exhibited modest increases in total antibiotic activity, as compared with the initial strains; no significant differences were observed in extracts from the biomass grown in liquid R5S (Appendix A). We picked two variants, further referred to as KO-1305 (a derivative of KO-1304, see Figure 1) and KO-1307 (a derivative of *rpsL* merodiploid lineage KO-1301), for further analysis. These strains maintained vigorous growth on solid media supplemented with MIC of erythromycin after five passages under nonselective conditions. However, KO-1304 and KO-1307 as well as wild type and KO-1304 exhibited the same erythromycin MIC in liquid culture (12 µg/mL). The KO-1305 and KO-1307 genomes carried *rpsL* and *rsmG* mutations inherited from their parents but also harbored a few novelties, as summarized in Table 1.

Despite the fact that both KO-1305 and KO-1307 were isolated in a single campaign and possess similar levels of erythromycin resistance, they do not share identical mutations, which might be responsible for their Ery^r^ phenotype when growing on solid media. Likewise, none of the revealed mutations/genes have obvious relations to antibiotic resistance, as detailed in the Discussion. Although only 0.2% of genomes of KO-1305 and KO-1307 lie in poorly sequenced regions, we could not rule out a possibility that some causative mutations escaped identification. For the moment, the exact nature of increased erythromycin resistance of aforementioned strains remains obscure.

### 3.2. Antibiotic Resistance and Bioactivities of Rifampicin-Resistant Derivatives of KO-1305 and KO-1307

The above-described Ery^r^ strains KO-1305 and KO-1307 were subjected to the next round of selection for increased rifampicin resistance (Rif^r^). The isolation of the 16 Rif^r^ mutants (KO-1401 to KO-1417), their Rif^r^ levels on solid media, and the identity of rpoB mutations have been reported previously [12]; the latter is also summarized in Appendix A. Here, we focus on the elucidation of pleiotropic effects that *rpoB* mutants might cause, such as altered resistance to other antibiotics and changes in specialized metabolism. We also carried out an in-depth genetic analysis of selected mutants.

All mutants grew vigorously on solid media and in liquid culture, although sporulation of some of the mutants on agar plates was retarded as compared with their parents (Appendix A). We profiled the resistance of Rif^r^ strains to novobiocin, thiostrepton, and ristomycin; the results are summarized in Table 2.

The antibiotics being tested represent mechanisms of action different from rifampicin and different between themselves, as mentioned below. This revealed that most of the mutants did not differ in their resistance to the aforementioned drugs as compared with their parents, while some strains become more susceptible. Particularly, strains KO-1403, 1406, 1407, 1408, and KO-1410 are more sensitive to more than one antibiotic. KO-1410 is unique in its high susceptibility to translation inhibitor thiostrepton and gyrase inhibitor novobiocin (KO-1410 was the only strain exhibiting significantly changed resistance to the latter drug). Cell wall inhibitor ristomycin was the drug to which the largest number of strains showed decreased resistance. Our data suggest that at least some of the studied *rpoB* mutations come at a notable fitness cost (in the form of increased drug susceptibility) for *S. albidoflavus*. Several mutants (those who showed altered resistance to the above-mentioned drugs and those who did not) were tested for their resistance to a wider set of antibiotics. We observed not only increased susceptibility to various drugs but also loss of susceptibility to cephalexin in strain KO-1414 (Appendix A).

Strains were grown on a set of solid media and their bioactivity (with an emphasis on an antibacterial one) was initially assayed with the agar plug method. The results, summarized in Table 3, demonstrated that certain mutants exhibited an enhanced ability to produce bioactive compounds in a medium-dependent manner. Strains KO-1401 and KO-1408 were particularly active in these tests. An example of plug assay is given in Appendix A.

To get more insight into the composition of specialized metabolites accumulated by the studied mutants, we grew them in liquid medium R5S and analyzed ethyl acetate extracts using disk diffusion and LC-MS approaches. The bioassay data for all strains and selected LC-MS results are shown in Figure 2, and all MS traces are presented in Appendix A.

Overall, our submerged fermentation-based results paralleled those observed for solid media. Namely, strains based on pure *rpsL* mutants were more potent than the KO-1307-derived lineage based on *rpsL* merodiploidy. Several strains, most notably KO-1401, showed strong bioactivity. The KO-1408-harboring Arg440→His substitution (one of the most frequently encountered variants among high-producing Rif^r^ mutants [24,25,26]) in this particular medium showed negligible antibiotic activity (see also Figure 2), although leading to increased bioactivity on solid media. Our observations underscore the fact that the observed biological activities stem from the combination of culture conditions and certain rpoB mutations. LC-MS data supported the idea that augmented antibacterial activity of the extracts of certain strains comes from an overproduction of paulomenols and paulomycins. This is the case for KO-1401 and KO-1416 (almost threefold increase for the former, as judged from mass peak areas), as well as other mutants, as detailed in Appendix A. No new mass peaks for any of the studied strains were found when growing them in R5S (see Figure 2 and Appendix A).

To reveal the impact of the fermentation medium on specialized metabolism, we extracted and analyzed small molecules produced by strains KO-1305 and KO-1408 in a liquid medium SG2. Here, we observed the increase in production of a few compounds, both observed before in *S. albidoflavus* as well as in unknown ones (Figure 3).

### 3.3. Heterologous Expression of Aranciamycin BGC in Selected Mutant Strains

Cosmid pOJara directing the biosynthesis of the aglycone of the anthracycline compound aranciamycin (aranciamycinone) was introduced into SAM2, KO-1305, KO-1307, and several Rif^r^ mutants to test their potential for heterologous production. Levels of aranciamycinone production by the aforementioned recombinant strains in a liquid medium are summarized in Figure 4. The three most productive strains, in descending order, were KO-1307, KO-1408, and KO-1403. None of the two tested Rif^r^ derivatives of KO-1307 exceeded the aranciamycinone productivity of the latter. Nevertheless, all studied mutants produced more aranciamycinone than their common ancestor strain SAM2.

### 3.4. Genomics of Strains KO-1403, KO-1407, KO-1408, KO-1412

The observed phenotypes may be a direct cause of the RifR mutations. However, it cannot be ruled out that unanticipated mutations outside of the *rpoB* locus are also involved. To address this issue, we sequenced and analyzed the genomes of several mutants. We focused on strain KO-1403 because its *rpoB* mutation was described for the first time in *Streptomyces* sp. (a summary of all known rpoB mutations can be found in Appendix A), as well as KO-1407, KO-1412 (rare *rpoB* mutations), and KO-1408 (known for favorable effects on specialized metabolism in other species [26]). No new mutations were revealed in KO-1408 and KO-1412 as compared with their parents, KO-1305 and KO-1307, respectively. A few mutations were identified within KO-1403 and KO-1407 genomes (Table 4), likely having a negligible impact on the physiology of their carriers (please see the Discussion).

### 3.5. Construction and Studies of S. albidoflavus rpoB Merodiploids

We cloned *rpoB* alleles from wild-type SAM2, KO-1403, and KO-1407 strains (*rpoB*^WT^, *rpoB*^S433W,^ and *rpoB*^R440C^, respectively) into the integrative vector pTES [23] under the control of the strong constitutive promoter *ermEp**. The resulting plasmids (pTESrpoB-WT, pTESrpoB-1403, and pTESrpoB-1407) were conjugally transferred into the initial strain of SAM2. While SAM2 transconjugants carrying plasmids pTESrpoB-WT and pTESrpoB-1407 exhibited a morphology that is typical for this species; the pTESrpoB-1403^+^ clones were of two different morphotypes, “typical” and “bald” (Appendix A), occurring on mating plates at a similar frequency. These morphotypes remained stable after five passages on solid media, so we picked both of them for further analysis. Thus, we generated the *rpoB* merodiploids SAM2-WT, SAM2-1403T (typical morphotype), SAM2-1403B (bald morphotype), and SAM2-1407. We used these merodiploids to address a few questions about the function of the cloned alleles. First, we wondered whether the presence of the Rif^r^-conferring *rpoB* allele in the background of wild-type *rpoB* would render the merodiploid more resistant to rifampicin. This was not the case: neither disk diffusion nor dilution spot assays revealed any differences in the resistance of SAM2 and the aforementioned merodiploids to two different concentrations (5 and 10 µg/mL) of rifampicin. Second, we wanted to find out if *rpoB* merodiploidy would impact specialized metabolism. SAM2-WT, SAM2-1403T, and SAM2-1407 merodiploids exhibited lower antibiotic activity on all tested solid media, while SAM2-1403B produced more antibacterials than SAM2 (see Appendix A).

## 4. Discussion

*S. albidoflavus* J1074 shows great promise as a platform for heterologous production of various small molecules that can used as drugs [3], nutraceuticals [27], biofuel components [28], and in other applications. The low yield of target molecules is a recurring problem of all heterologous expression approaches, requiring tedious optimization of the chassis strains and/or production process [29,30]. It is of interest, therefore, to identify genes/mutations whose introduction would have a global favorable effect on the specialized metabolism of *Streptomyces*. Encouraged by the observation of such effects as a result of point mutations in gene *rpsL* for ribosomal protein S12 [9], we went on to generate multiple drug-resistant derivatives on the basis of *rpsL* mutants. In this report, we describe the properties of two groups of such mutants, sequentially selected for increased resistance to erythromycin and rifampicin, representing a pinnacle of a long selection campaign as depicted in Figure 1.

The molecular basis of the increased resistance of KO-1305 and KO-1307 to erythromycin remains unknown. These strains have no mutations known to cause erythromycin resistance [31,32]. Likewise, we did not reveal mutation(s) common to both strains, which would underlie their resistance phenotype. KO-1305 harbors a missense mutation in gene *xnr_4320* coding for a putative Na^+^/H^+^ antiporter, which leads to Trp370→Arg substitution in the last cytoplasmic segment of this protein. Gene *xnr_4895* encodes a 286-aa transcriptional factor of an unknown function with an N-terminal DNA-binding HTH domain. A single-nucleotide deletion at the 113th position (G) of the gene will result in the production of an altered and truncated (62 aa) protein that appears to retain its functional HTH domain. In KO-1307, three missense mutations were revealed. Gene *xnr_3039* for the ortholog of cell division protein FtsW (474 aa) carries a substitution within the second position of the 154th codon, leading to an Ile→Ser change. According to a transmembrane topology prediction using TMHMM (https://services.healthtech.dtu.dk/service.php?TMHMM-2.0 (accessed on 1 February 2023)), Ile154 is located on the cytoplasmic side of FtsW, at the very start of its fifth transmembrane helix. Gene *xnr_3154* carries a mutation that leads to conservative Ile→Leu substitution in the C-terminal part of the 1211-aa LamG/jellyroll domain protein Xnr_3154 of unknown function. Finally, a mutation within the gene *xnr_4783* for putative transglutaminase (808 aa) causes Leu757→Pro substitution. None of the abovementioned mutations/genes fit the straightforward hypothesis of their involvement in erythromycin exclusion or modification. Further experimental scrutiny is needed to establish their association with macrolide resistance.

We were able to isolate a wide variety of *rpoB* variants in KO-1305 and KO-1307, of which one, to the best of our best knowledge, KO-1403 (*rpoB*^S433W^), was isolated for the first time. Some of the *rpoB* mutations influenced antibiotic resistance and the specialized metabolism of *S. albidoflavus*. Strains KO-1401, KO-1403, KO-1404, and KO-1408 were particularly productive under different cultivation conditions. The effects of *rpoB* mutations on the ability of *S. albidoflavus* to produce specialized metabolites are worth a few comments. First, the genetic background of KO-1305 (pure *rpsL* lineage) was more favorable for endogenous specialized metabolism than that of KO-1307. A comparison of the bioactivities of strains KO-1404 and KO-1411 harboring the same *rpoB*^H437R^ mutation (see Figure 2) provides ample illustration of this point. Second, the type of growth medium and mode of cultivation strongly impact the antibiotic activity of the mutants, implying that some of the low-activity strains might exhibit greater promise under different conditions, as often has been reported in the recent literature [33]. In the vein of these observations come also reports on the effects of nutritional status on antibiotic resistance, which should be taken into account [34]. Third, successful heterologous expression would also require finding the optimal combination of a chassis genotype and production conditions for a given biosynthetic pathway. Indeed, none of the *rpoB* mutants exceeded the aranciamycinone productivity of KO-1307, whose endogenous specialized metabolism showed no significant improvement as compared with parent strains and its Rif^r^ derivatives. Fourth, it appears that the enhanced specialized metabolism of Rif^r^ strains comes from the introduction of the *rpoB* mutation and not an unanticipated genetic variation. After sequencing four strains, we revealed additional mutations outside of the *rpoB* locus in KO-1403 (1 nonsense and 1 missense mutation) and KO-1407 (1 substitution). Only one of these mutations occurs in a gene of potential relevance (*xnr_4479* for putative two-component sensor histidine kinase), albeit it is a conservative Ala→Val substitution. Fifth, our pilot study of two Rif^r^-conferring *rpoB* alleles in the background of wild-type *rpoB* shows that merodiploids exhibit altered endogenous specialized metabolism. This justifies a wider investigation of the mechanisms and consequences of *rpoB* merodiploidy. To conclude, our collection and characterization of drug-resistant *S. albidoflavus* strains can be used to identify optimal host(s) for the expression of BGC of interest.

## Figures and Tables

**Figure 1 microorganisms-11-01176-f001:**
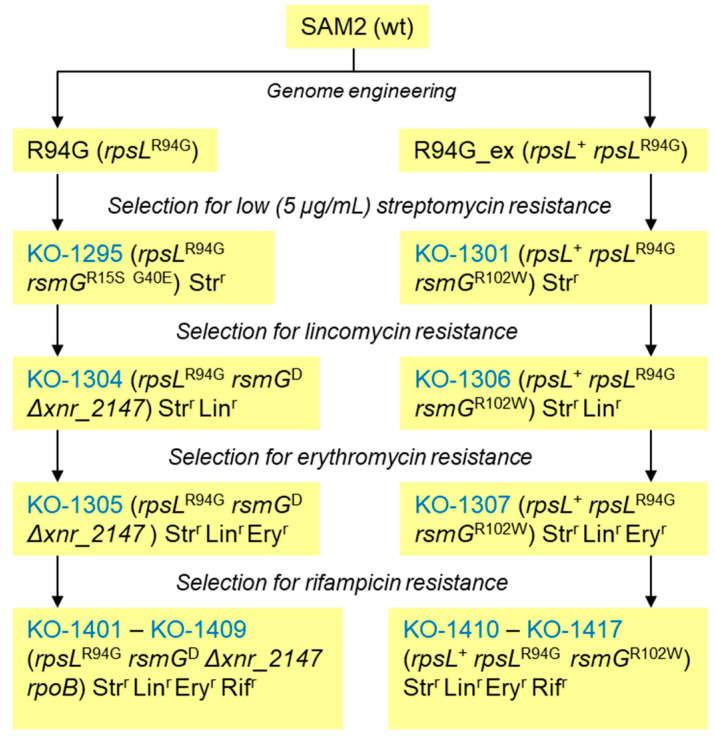
The pedigree of *Streptomyces albidoflavus* J1074 strains described in this work (adapted from [12]). SAM2 is a derivative of J1074, carrying the deletion of the pseudo *attB*^φC31^ site [10]. R94G is an initial pure *rpsL* mutant that served to generate multiple resistant spontaneous mutants. R94G_ex represents *rpsL* merodiploid lineage. Mutants KO-1401 to KO-1417 (bottom of the figure) are the focus of this study; they represent the final stage of the strain selection where *rpoB* mutations were introduced. Identities of lincomycin resistance mutations in KO-1306 and erythromycin resistance determinants in KO-1305 and KO-1307 remain unknown (see also main text). Names of spontaneous mutants are shown in blue. Mutant allele *rsmG*^R15S G40E^ (see KO-1295) in the derivative strains is labeled as *rsmG*^D^ for the sake of brevity.

**Figure 2 microorganisms-11-01176-f002:**
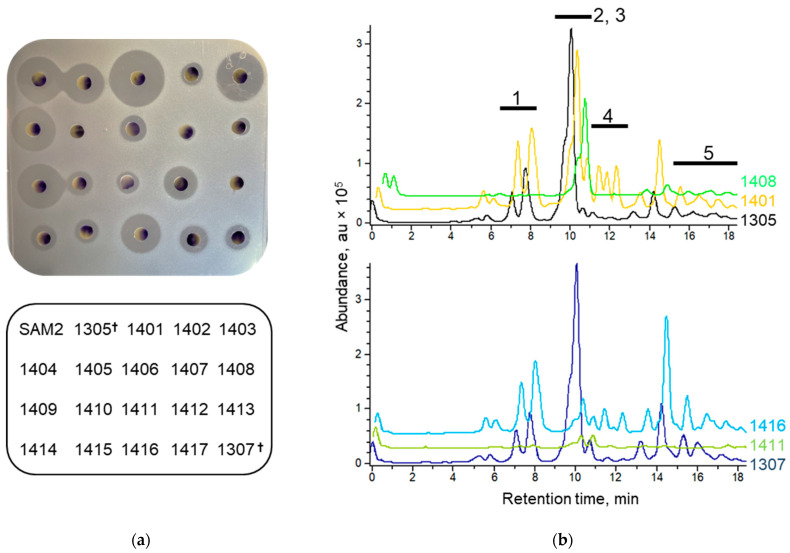
Endogenous specialized metabolism of selected mutants. (**a**) Results of disk diffusion assay of extracts from R5S-grown strains against *B. cereus*. The photograph represents the typical result of at least three biological repeats. The graphical legend below the photograph denotes strain extracts on the plate. Dagger sign marks initial strains. (**b**) Overlaid LC-MS traces from the extracts are denoted to the right of the chromatogram. The traces are normalized to equal amounts of biomass. Bold black lines denote fractions of known natural products: 1, paulomenols; 2 and 3, surugamides and candicidines; 4, paulomycins; and 5, antimycins.

**Figure 3 microorganisms-11-01176-f003:**
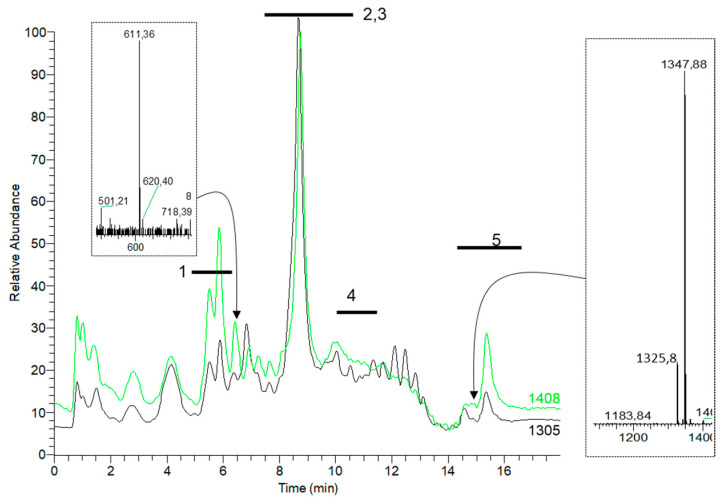
Endogenous specialized metabolism of strains KO-1305 and KO-1408 grown in medium SG2. Overlaid MS traces are normalized to the biomass and labeled with thick horizontal lines denoting known families of small molecules (1–5; see the legend in Figure 2) produced by this species. Please note that for this experiment we used a shorter LC separation program; hence, there are shifts in compounds retention time as compared with Figure 2. Insets (dotted rectangles) denote two unknown mass peaks: 611.4 Da present in KO-1408 in 10-fold excess (as compared with KO-1305), and 1325.8 (and its sodium adduct, 1347.8 Da) Da compound found only in KO-1408 extracts. This MS trace chromatogram represents a typical result of four biological repeats.

**Figure 4 microorganisms-11-01176-f004:**
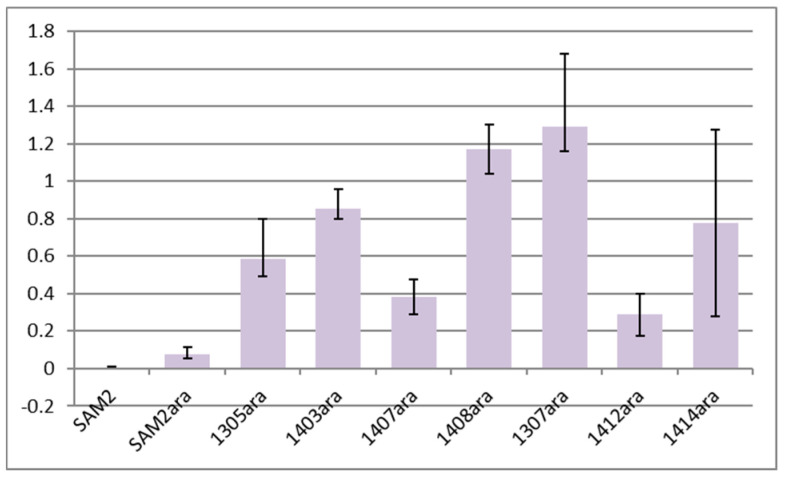
Production of aranciamycinone by drug-resistant S. albidoflavus pOJara^+^ strains in liquid medium SG1. Production was normalized to equal amount (10 mg) of dried biomass. Results represent mean values of four repeats; error bars represent ±2SD.

**Table 1 microorganisms-11-01176-t001:** Novel mutations within KO-1305 and KO-1307 genomes as compared with their parents.

Gene/Region (Function)	KO-1305	KO-1307
*xnr_3039* (FtsW, cell division)	- ^1^	3455194; I→S
*xnr_3154* (LamG/jellyroll domain protein)	-	3594936; I→L
*xnr_4320* (Na^+^/H^+^ antiporter)	4919731; W→R ^2^	-
*xnr_4895* (Transcriptional factor)	5569092; ∆T ^3^	-
*xnr_4783* (Transglutaminase)	-	5437784; L→P

^1^ No mutation; ^2^ position and type of substitution; ^3^ deletion site.

**Table 2 microorganisms-11-01176-t002:** Results of antibiotic disk diffusion assay of resistance of Rif^r^ strains to thiostrepton, novobiocin, and ristomycin.

*S. albidoflavus* Strain	Growth Inhibition Zones (mm)		
	Thiostrepton (50 µg/disk)	Novobiocin (50 µg/disk)	Ristomycin (50 µg/disk)
SAM2	23 ± 2	30 ± 3	22 ± 2
KO-1305 ^1^	28 ± 3	30 ± 2	28 ± 2
KO-1401	27 ± 2	27 ± 2	25 ± 2
KO-1402	34 ± 2	26 ± 1	31 ± 3
KO-1403	35 ± 2 ^2^	38 ± 2	40 ± 5
KO-1404	26 ± 2	30 ± 2	27 ± 2
KO-1405	32 ± 2	33 ± 2	28 ± 3
KO-1406	35 ± 2	34 ± 2	40 ± 3
KO-1407	33 ± 2	37 ± 2	34 ± 2
KO-1408	35 ± 2	35 ± 2	36 ± 3
KO-1409	21 ± 2	27 ± 2	26 ± 1
KO-1307 ^1^	32 ± 3	28 ± 2	31 ± 3
KO-1410	35 ± 2	42 ± 2	42 ± 2
KO-1411	30 ± 2	29 ± 2	32 ± 2
KO-1412	26 ± 2	35 ± 2	33 ± 2
KO-1413	28 ± 2	30 ± 2	28 ± 1
KO-1414	23 ± 1	27 ± 1	22 ± 1
KO-1415	32 ± 2	38 ± 1	40 ± 2
KO-1416	28 ± 2	35 ± 3	30 ± 3
KO-1417	28 ± 2	31 ± 3	27 ± 3

^1^ Parental strain for respective Rif^r^ lineage. ^2^ Values on grey background are significantly different from the parental strains (the difference between maximal and minimal values of growth inhibition zones of parental and mutant strains, respectively, are more than 5 mm).

**Table 3 microorganisms-11-01176-t003:** Antibiotic activity of Rif^r^ mutants grown on agar media.

Medium	SG2		SFM	TSA	GYM		MM
Strain	*B.c.* ^1^	*D.h.* ^2^	*B.c.*	*B.c.*	*B.c.*	*S.a.* ^3^	*B.c.*
SAM2	+++ ^4^	++	+	+	+	+	+
KO-1305	+++	+	–	–	+	++	–
KO-1401	++++	++	ND ^5^	ND	ND	+++	ND
KO-1402	+++	++	ND	ND	ND	++	ND
KO-1403	+++	++	–	–	++	+++	–
KO-1404	+++	++	ND	ND	ND	+++	ND
KO-1405	+++	+++	ND	ND	ND	++	ND
KO-1406	+++	++	–	–	++	++	+
KO-1407	+++	+++	+	++	–	+++	++
KO-1408	++++	+	+	++	–	+++	++
KO-1409	+++	++	ND	ND	ND	++	ND
KO-1307	++	++	–	–	+	+	ND
KO-1410	+	+++	ND	ND	+	++	ND
KO-1411	–	+	ND	ND	+	++	ND
KO-1412	++	+++	ND	ND	++	++	ND
KO-1413	–	+	ND	ND	–	++	ND
KO-1414	++++	+	ND	ND	+	++	ND
KO-1415	+++	+	ND	ND	+	++	ND
KO-1416	++	++	ND	ND	++	+++	ND
KO-1417	++	++	ND	ND	+	++	ND

^1^ *B.c*., *Bacillus cereus*. ^2^ *D.h.*, *Debaryomyces hansenii*. ^3^ *S.a.*, *Staphylococcus aureus* 209P. ^4^ Antibiotic activity level: –, no activity; +, diameter of growth inhibition halo (Ø) is from 7 to 10 mm; ++, Ø is from 11 to 15 mm; +++, Ø is from 16 to 20 mm; ++++, Ø is equal to or more than 21 mm. Results are mean values of three biological repeats; deviations constitute no more than 10% of the mean. ^5^ ND, not determined.

**Table 4 microorganisms-11-01176-t004:** Novel mutations within KO-1403 and KO-1407 genomes as compared with their parents.

Gene/Region (Function)	KO-1403	KO-1407
*xnr_1600* (secreted protein)	1,896,611; Y→UAG ^1^	-
*xnr_2792* (unknown)	-	3,168,482; S→G
*xnr_4479* (TCS ^2^ histidine kinase)	5,095,310; A→V	-

^1^ See Table 1 for explanation of abbreviations and notations. ^2^ Two-component sensor system.

## Data Availability

All data supporting the results of this study can be found in Appendix A and links provided in Section 2. The mutants and plasmids described in this study are available from B.O. and A.L. upon reasonable request.

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
