# Peer review of "Properties of Multidrug-Resistant Mutants Derived from Heterologous Expression Chassis Strain Streptomyces albidoflavus J1074"

_microorganisms, 2023, doi:10.3390/microorganisms11051176_

Round 1

Reviewer 1 Report

Streptomyces albidoflavus J1074 is widely used as a host for heterologous expression of biosynthetic gene clusters (BGCs). Although rpoB mutations are known to be effective to activate expression of cryptic secondary metabolite-BGCs, the effects on this strain remained unstudied. In this study, the authors isolated rifampicin-resistant strains of J1074 and consequently obtained 14 rifampicin-resistant strains with mutated rpoB. By their characterization, it is concluded that rpoB mutants are useful to enhance the ability of J1074 to produce secondary metabolites.

As this study is significant for readers and well done. The manuscript is also well written. I recommend it to be accepted after addressing minor points shown below.

1) Please clarify the relationship between SAM2 and J1074 although it may be described in reference 12.

2) L82. ATCC19637 -> ATCC 19673 (please insert a single space between ATCC and the strain number)

3) L132. ‘55÷165’ means 0.333. The coverage of 0.33-fold is too low.

4) L144 and L146. Is ‘KO-1301’ correct? I think it may be typo of KO-1306.

5) Tables 2 and S2. Antibiotics -> Growth inhibition zone (mm)

6) Table S2. Please describe the amount of antibiotics attached to paper disks in the footnote.

Author Response

I thank for favorable comments on our work. Below are point-by-point answers to the comments.

  1. The relationship between SAM2 and J1074 is clarified in the first line of the legend of Fig. 1. More information is in refs 10 and 12, to which we refer an interested reader
  2. Done, thank you
  3. Corrected; we implied that coverage is from 55- to 165-fold
  4. Corrected, thank you.
  5. Corrected, thank you.
  6. The information on antibiotic amount is now provided in footnote, see the revised supplementary information

Reviewer 2 Report

The work by Dolya and co-workers provides insights about three different features that are not necessary interconnected (antibiotic resistance, production of bioactive compounds and the enhancing of the heterologous expression properties of the strains), which makes difficult to understand the specific aim of the work. Nevertheless, the results are sound and the techniques performed appropriated. I think the work can be organized better so as to clearly indicates what is the final aim of the study.

In addition, I have some specific comments:

·      I do not understand the selection of the grey boxes of Table 2 in the thiostreptone resistance. There are some values that are also significantly different (i.e. KO-1409, KO-1412, KO-1414, etc.) In other words, why the strains that behave more resistant to the antibiotic in comparison to its respective parental strain are not shown in grey nor discussed?

·        In the legend of Figure 2 the indicator strain (to which the antagonistic assay has been performed) should be mentioned.

·        Why for the LC-MS analysis is used R5S medium when in the assays with solid media the R5 solid medium was not used? It is no obvious the reason behind.

·        Why the bioactivity against the yeast (Debaryomyces hansenii) was only evaluated on SG2 medium?

·        Was the bioactivity evaluated against Gram-negative bacteria and other fungi species (for example Candida albicans).

·        Pictures of the bioassays on solid media should be included in supplementary material and also a couple of examples of the antibiotic disk diffusion assays with the distinct antibiotics mentioned in the strains selected.

·        The XNR_2147 deletion of the parental strain KO-1304 should be discussed in the context of the results shown in Table 1. I cannot see the mutation in the text.

·        Results of the LC UV spectra of the figure 2B should be better explained in the context of the results of the Mass Spectrometry (MS).

Minor corrections:

-        L37-38. “and was also used for the incorporation of nonproteinogenic amino acids into ribosomally produced [4]”. This sentence seems incomplete.

-        I cannot see the connection of the subheading “3.2. Antibiotic resistance and bioactivities of rifampicin-resistant derivatives of KO-1305 and 169 KO-1307 as revealed by bioassays and LC-MS”. I would first described the resistance patterns of the strains, and in other section, the bioactivity of the strains against the different microorganisms.

-        Legend of Fig. 2B. Clarify: “Overlaid MS traces..”Could it be LC-MS?

-        L335-337: “Second, type of growth medium and mode of cultivation strongly impact antibiotic activity of the mutants, implying that some of the low activity strains might exhibit great promise under different conditions”. Include and discussed the work: PMID: 35185841 (Identification of Antimicrobial Compounds in Two Streptomyces sp. Strains Isolated From Beehives)

-        The same applies for the nutritional regulation observed for the antibiotic resistance profiles of the strains on the experiment shown in Table 3. Include and discussed the work: PMID: 29898657 (Genome sequencing analysis of Streptomyces coelicolor mutants that overcome the phosphate-depending vancomycin lethal effect)

Author Response

We thank for favorable review of our work. First of all, I note that, as title of our manuscript states, we focus on properties of the mutants not limiting to their certain kind, because rpoB point mutations exert pleiotropic effects. As unconnected these properties may seem, they all stem from the change in rpoB. Thus our aim was to study all the diversity of effects that rpoB, in the background of the other mutations, may cause to J1074. This point is mentioned in the text - see L. 175, for example.  Below are the answers to specific comments.

  1. The Table 2 is now corrected. We focused on significant differences as they are explained in the footnote, and now these differences are highlighted with grey background correctly. We also re-write the discussion of these results to include all observed differences - see LL. 192-198.
  2. The Figure 2 legend is now amended to mentioned the indicator strain.
  3. We used solid R5S medium, as mentioned in the Fig. 2 legend, there is a typo in Methods. It is now corrected. We used R5S because it is simpler than R5 and showed excellent results for S. albidoflavus.
  4. We were mostly interested in finding the conditions that elicit antimicrobial activity. S. albidoflavus is known to produce at least four classes of antifungals - surugamides, alteramides, candicidins and antimycins. There is plenty of antifungal acitivity and it is no surprise to observe it. Therefore we focused more of antibacterial potency. We now add a line about it in the main text, see L. 204.
  5. In this study we used only the indicator strains mentioned in the text.
  6. We did not take photos of all antibiotic resistance and bioactivity assays, only measured the diameters of growth inhibition. Some photos are available, and we now show them in Suppl Mat - pelase see the bottom of Table S2, as well as new Fig. S5 (agar plug assay).
  7. The mutation in Ko-1304 is not present in the Table 1, as this strain has alredy been described and published - please see the Introduction and ref 13 mentioned there. In table 1 we focus on new mutants 1305 and 1307.
  8. The Fig. 2B shows LC-MS spectra, not UV, it is mentioned in the legend. We discuss these results below the figure, see LL. 234-237.

Minor corrections

  1. Corected, thanks - word "peptides" added
  2. We thank for this comment. Indeed, title sounds awkward, I have revised it - pelase see the new version. I do prefer to described everything we know about these strain in a single section.
  3. OK - now it is LC-MS traces
  4. OK - included
  5. OK - phosphate paper included as well